# Exploring the Relationship Between Clinical Supervision and Well-Being in the Otolaryngology Residency Board in Saudi Arabia

**DOI:** 10.3390/healthcare13030328

**Published:** 2025-02-05

**Authors:** Mohammad Ali Alessa, Sarah Ahmed Eltouny, Hashem O. Alsaab, Rabab Abdel Ra’oof Abed

**Affiliations:** 1Head and Neck & Skull Base Health Centre, King Abdullah Medical City, Makkah 24246, Saudi Arabia; 2Pharmaceutical Sciences Department, Fakeeh College for Medical Sciences, Jedda 23323, Saudi Arabia; 3Faculty of Medicine, Suez Canal University, Ismailia 41522, Egypt; sarah.eltony@gmail.com (S.A.E.); dr.rababraoof@gmail.com (R.A.R.A.); 4Department of Pharmaceutics and Pharmaceutical Technology, Taif University, Taif 21944, Saudi Arabia

**Keywords:** surgical residency, mental well-being, clinical supervision, otolaryngology training, resident burnout

## Abstract

Background/ Objectives: Surgical residency is widely recognized as a highly stressful phase due to long working hours and the challenges of managing complex cases. Additionally, family responsibilities, such as being a spouse or parent, can have a positive or negative impact on residents’ well-being. This study aimed to explore the relationship between clinical supervision and mental well-being among otolaryngology residents in Saudi Arabia, focusing on how supervision conditions influence well-being at different stages of training. Methods: This was an analytical cross-sectional correlational study conducted among Saudi otolaryngology head and neck surgery residents. An online survey was used to collect data from 64 residents, utilizing the Dutch Residents Educational Climate Test (D-RECT) to assess clinical supervision and the Warwick–Edinburgh Mental Well-being Scale (WEMWBS) to measure well-being. The data were analyzed to determine the association between supervision conditions and well-being across different residency levels. Results: The results showed that the majority of residents reported higher mean scores for items such as “I’ve been feeling useful” (3.53 ± 1.23), “I’ve been feeling interested in new things” (3.28 ± 1.13), and “I’ve been dealing with problems well” (3.27 ± 1.10). No statistically significant difference in overall WEMWBS scores was found between junior and senior residents. However, mental well-being was significantly associated with all four D-RECT domains (supervision, feedback, coaching assessment, and consultant attitude), with a positive correlation observed between clinical supervision and well-being. Conclusions: This study highlights the critical role of clinical supervision in supporting the mental well-being of otolaryngology residents. Enhanced supervision practices, particularly those emphasizing constructive feedback and supportive consultant attitudes, could mitigate burnout and improve resident outcomes. These findings underscore the need for targeted interventions in residency programs to promote well-being and optimize the learning environment.

## 1. Introduction

A core mission of residency training programs is to develop a learning environment that facilitates the safe development of clinical skills and knowledge to prepare trainees for future independent practice [1]. Residents face various stressful challenges, including heavy workloads and difficulties in balancing work and social life owing to long working hours [1]. Such work-related factors could affect mental well-being and ultimately lead to burnout or mental illness, such as depression or anxiety disorders.

Clinical supervision was first developed by Cogan in 1973 after realizing the significance of interactions among clinical teachers and learners to promote professional advancements [2]. The American Medical Association and College of Surgeons conducted research in 2017 to determine the standards of medical trainees’ specialties concerning the level of training and supervision from medical experts. According to the findings, clinical supervision highly improved learners’ cognitive and professional skills in medical fields, thus enhancing the quality of services offered [3]. Boor et al. showed that supervision is the most important factor that leads to high-quality residency training [4]. Clinical supervision plays an important role in reducing burnout among residents when there is a supportive relationship between residents and their supervisors; on the other hand, a stressful relationship is associated with increased levels of burnout among trainees [5].

Clinical supervision is conceptualized as a critical job resource that mutually reinforces personal skills and promotes well-being [2]. Thus, appropriate supervision in healthcare settings is shaped by the factors that influence job well-being and, in turn, impact those outcomes.

Mental well-being refers to the state of being in control of the emotional, physical, and psychological constituents that actively and passively affect one’s quality of life. Being in a state of emotional stability and acknowledging that stressors can be managed or averted are practical models that explain mental well-being [6]. Furthermore, a person’s mental health could reflect his/her mental well-being [7]. According to a holistic approach, some studies believe that mental well-being reflects life’s stressors and the foreseeable ability to address these emerging or existential issues [6].

Some may think that burnout and well-being are two faces of the same coin or may define well-being as the absence of burnout [8]; however, there is a difference between the two concepts. Well-being is a complex psychological construct that consists of multiple factors. Ryff developed a scale that is used for measuring well-being. This scale is composed of six factors that identify well-being, namely, self-acceptance, autonomy, purpose in life, environmental mastery, personal growth, and positive relationships [9].

Two distinct approaches define well-being: the hedonic and eudaimonic approaches. The hedonic approach defines well-being as subjective happiness and judgments regarding good elements of life. However, happiness does not relate solely to physical hedonism but could relate to the attainment of goals. The eudaimonic perspective assumes that satisfying desires does not guarantee well-being, as some personal goals are not good for people. Eudaimonia emphasizes that well-being can be achieved when there is congruence between life activities and deeply held values (true self) and is more related to self-realization and actualization [10].

Mental well-being and clinical supervision during residency training have vital impacts on healthcare providers’ performance as well as on the population that they serve. Clinical supervision has negative repercussions that lead to worker burnout, a common syndrome of exhaustion [11]. Empirical findings suggest that many trainees’ well-being is affected by experiencing emotional drain, depersonalization, and lack of personal accomplishment regarding the individual’s professional activity [3]. This can be attributed to excessive pressure from clinical supervisors, heavy workloads, and high levels of strictness in accordance with clinical settings.

Residency training and clinical supervision focus on the workplace, reflecting on issues such as burnout in the workplace. Burnout in the workplace results in a lack of mental well-being. Chronic stress in the workplace causes burnout. According to Rothenberger (2017), once the workplace is properly managed, burnout can be controlled) [7]. The stressors that emanate from the workplace cause burnout, but the chronic form of burnout one feels can easily lead to a lack of mental wellness. When people suffer from burnout for a long period of time, they become exhausted physically, emotionally, mentally, and psychologically [12]. A stressed person is exposed to overload syndrome, which predisposes him/her to compromised mental well-being.

This study aims to explore the relationship between clinical supervision and the abilities, skills, and performance quality of medical trainees, emphasizing the importance of fostering effective learner-supervisor interactions in the medical learning environment [3]. Through study, society will realize and appreciate the quality of care and treatment provided by healthcare providers as they believe in their expertise and specialty [1]. Based on the evaluation of the discussed issue from different resources of literature and empirical frameworks, this study aimed to assess the relationship between clinical supervision and well-being via the Otolaryngology Residency Board, thus providing valuable insight into residents’ perceptions and welfare regarding their level of care. Furthermore, this research pays close attention to measuring the state of well-being in otolaryngology head and neck surgeons residents in Saudi Arabia. The results of hospital operational strategies vary depending on clinical supervision. The efficacy of the clinical supervisor and the clinical care itself seems to influence the trajectory of this outcome [13]. This study emphasizes the necessity for businesses to invest in high-quality methods of supervision to achieve the most benefits from clinical governance.

## 2. Materials and Methods

### 2.1. Research Design

This was a descriptive cross-sectional study conducted among Saudi national otolaryngology resident board members. A comprehensive sampling technique was used, where all the residents from all over the Kingdom of Saudi Arabia (KSA) who were in their second to fifth years of residency were involved to ensure enough exposure to clinical supervision in each center, as rotation changes every six months. Residents in Level 1 were excluded because their otolaryngology rotation was less than six months, and they may have rotated into tertiary hospitals that were outside the scope of the study. The study was conducted through an online questionnaire distributed by chief residents and the Program Director of each region in the country; data were collected over an 8-week period.

Sixty-four postgraduate residents participated in the study and completed the questionnaire. 

### 2.2. Data Collection Tools

A pretested and validated questionnaire was used to collect the residents’ responses. An online survey was created by combining D-RECT 4, which was developed for measuring clinical supervision. D-RECT consists of 50 questions in 11 dimensions. We selected four areas that are relevant to the study variables, namely, supervision, feedback, coaching assessment, and consultant attitude. The Warwick–Edinburgh Mental Well-being Scale (WEMWB) 11 was developed to measure well-being. It consists of 14 positively worded items to assess psychological functioning and subjective well-being. The score is measured by summing up responses on a 1–5 Likert scale and the maximum score is 70, while the minimum score is 14.

### 2.3. Ethical Consideration

Ethical approval was obtained from the Institutional Review Board (IRB) of Fakeeh College for Medical Science, Jeddah. The participants were assured of confidentiality, privacy, and the right to withdraw from completing the survey, with no effect on clinical training or grading.

### 2.4. Data Analysis

Statistical analysis was performed with the Statistical Package for Social Sciences version 21.0 for Windows (SPSS, Inc., Chicago, IL, USA). Descriptive statistics were used to present the participants’ characteristics and the patterns of answers given to the different questionnaire sections. Categorical variables are presented as frequencies and percentages, while continuous variables are presented as the means ± standard deviations (SDs). The internal consistency of all the questionnaire scales was analyzed by calculating Cronbach’s alpha, where values > 0.7 indicated the reliability of the given scale. Furthermore, the normality of the distributions of the different scores was tested by analyzing the distribution histogram and using the Kolmogorov–Smirnov and Shapiro–Wilk tests. The comparison of WEMWBS overall and item scores across the factor categories was analyzed using an independent t-test and one-way ANOVA; the results are presented as the mean (SD) for overall and item scores. Multivariate linear regression was used to analyze the independent association of clinical supervision parameters with the overall WEMWBS score using two methods: the enter and stepwise methods. The results are presented as unadjusted regression coefficients (B) with 95% confidence intervals (95% CIs). A *p*-value of <0.05 was considered to indicate statistical significance, and the null hypothesis was rejected.

## 3. Results

### 3.1. Participant Characteristics (Demographic Data)

Sixty-four postgraduate residents replied to the questionnaire.

### 3.2. Assessment of Mental Well-Being

The patterns of answers to the 14 items of the WEMWBS are depicted in Table 1. These items had higher mean (SD) scores for the following items: “I’ve been feeling useful” (3.53 [1.23]); “I’ve been feeling interested in new things” (3.28 [1.13]); and “I’ve been dealing with problems well” (3.27 [1.10]). The lowest mean (SD) scores were observed for “I’ve been feeling relaxed” (2.78 [1.17]), “I’ve had energy to spare” (3.03 [1.22]), and “I’ve been feeling interested in other people” (3.05 [1.17]).

The internal consistency of the WEMWBS had a Cronbach’s alpha as high as 0.958, indicating the excellent reliability of the answers. The mean (SD) WEMWBS score was 44.39 (12.95) out of 70 (range = 15–68) (Table 2).

### 3.3. Internal Consistency of the D-RECT Subscales and the WEMWBS

The supervision, feedback, coaching assessment, and consultant attitude subscales showed good to excellent reliability, with Cronbach’s alpha values of 0.740, 0.829, 0.912, and 0.928, respectively (Table 2). The four D-RECT scores were divided into two levels, satisfactory (score > 3 out of 5) and unsatisfactory (score ≤ 3 out of 5), and were analyzed as dichotomous variables. The satisfaction rates in the four domains were as follows: supervision (62.5%), feedback (40.6%), coaching assessment (60.9%), and consultant attitude (78.1%). The WEMWBS score demonstrated high internal consistency with Cronbach’s alpha (0.958), indicating the validity of this tool for evaluating mental well-being in Saudi residents.

### 3.4. Levels of Mental Well-Being by Residency Level

The mean (SD) WEMWBS item scores were not significantly different between junior (second and third years) (total score of WEMWBS 41.33) and senior (fourth and fifth) residents (total score of WEMWBS 45.88), except for the last item, “I’ve been feeling cheerful”, where the mean (SD) score was lower for junior (2.81 [0.98]) than for senior (3.40 [1.12]) residents (*p* = 0.045) (Table 3).

### 3.5. Factors Associated with the WEMWBS Score

There was no significant difference in the mean (SD) WEMWBS score according to residency year (*p* = 0.127). However, mental well-being was significantly associated with all four D-RECT score levels; that is the mean (SD) WEMWBS score was significantly higher in cases of resident’s satisfaction with supervision (48.48 [12.94] vs. 37.58 [9.91], *p* = 0.001), feedback (51.65 [10.04] vs. 39.42 [12.45], *p* < 0.001), coaching assessment (49.23 [11.11] vs. 36.84 [12.14], *p* < 0.001), and consultant’s attitude (48.20 [10.64] vs. 30.79 [11.42], *p* < 0.001) (Table 4).

Adjusted linear regression, using both enter and stepwise methods, showed that feedback and the consultant’s attitude were the only significant predictors of mental well-being. According to the results of the enter method, feedback was associated with an unadjusted regression coefficient B of 6.44 (95% CI = 0.49–12.39), while consultant attitude was associated with B = 11.21 (95% CI = 3.46–18.96); the model explained 38.7% of the variance in the WEMWBS score. According to the stepwise method, feedback had B = 7.98 (95% CI = 2.40–13.57), while consultant’s attitude had B = 13.99 (95% CI = 7.36–20.63); the model explained 37.5% of the variance in the WEMWBS score (Table 5).

## 4. Discussion

The main focus of this study is to look in depth at the impact of clinical supervision on mental well-being and to measure the degree of well-being from one level to another during training programs.

Regarding levels and patterns of mental well-being, the items that were associated with the highest levels of satisfaction were the following: “I’ve been feeling useful”, “I’ve been interested in new things”, and “I’ve been dealing with problems well”. These answers may be correlated with greater feelings of self-confidence and professional value, as well as positive feedback in the working environment. Furthermore, these dimensions can be linked to or reinforced by supervisors’ compliments and appreciation statements. In contrast, the three items “I’ve been feeling relaxed”, “I’ve had energy to spare”, and “I’ve been feeling interested in other people” had the lowest mean (SD). These expressions may be related to feelings of job comfort and relaxation, which are more strongly perceived in the context of satisfaction with one’s job ease or reduced work exhaustion.

Similarly, a Chinese study among university students had the highest mean WEMWBS item score for the item “I’ve been interested in new things”, followed by the two statements “I’ve been feeling loved” and “I’ve been feeling cheerful” [14]. However, the variance in the item scores from the latter study was relatively lower than that observed in our study, indicating greater disparity in mental well-being among our residents.

Lindemann et al. used the WEMWBS to evaluate the mental well-being of the next generation of general practitioners and showed that half of the participants had a high likelihood of burnout [15]. A study from Pakistan used the WEMWBS to assess the psychological impact of the COVID-19 pandemic on physicians’ mental health and reported emotional exhaustion in almost half of the participants and depersonalization in most of them, while almost three of them reported low personal accomplishment.

For the factors associated with the WEMWBS score, the current study findings showed that mental well-being was significantly associated with all four D-RECT items of supervision, feedback, coaching assessment, and consultant attitude. Consistently, a survey from the UK showed that doctors’ satisfaction was associated with strong clinical supervision, frequent and useful feedback meetings, adequate workload, and supportive environment [16]. In addition, emerging data reveal that the behavior of supervisors is crucial for trainees’ well-being and training satisfaction [17,18]. This may have a significant impact on the mental well-being of the trainee. A study from the Mayo Clinic reported that a 1-point increase on a 60-point scale of leadership in a physician’s immediate supervisor caused a 3.3% decrease in susceptibility to burnout (*p* < 0.001) and an increase in satisfaction of 9.0% (*p* < 0.001) [19]. In the present study, the mean (SD) WEMWBS score was significantly greater for residents’ satisfaction with feedback. Moreover, an online group coaching program significantly reduced burnout as well as emotional exhaustion and imposter syndrome among female resident physicians, whereas it increased their self-compassion [20].

Regarding the impact of the learning climate on residents’ well-being, feedback and the consultant’s attitude were the only significant predictors of mental well-being in the present study. These two factors reflect the sense of communication and warm relationships between trainees and seniors and have been shown to promote mental health in doctors [21]. Positive feedback and attitudes for young doctors and students, when the latter make mistakes or perform incorrectly, increase the student’s willingness to learn and outcomes [21]. In contrast, in the absence of feedback, trainees’ skills may not evolve, and their mistakes, knowledge gaps, and weaknesses might remain uncorrected [22]. Thus, feedback can maximize learning effectiveness by informing students about their progress and needs for improvement as can encouraging them to engage in appropriate learning activities [23]. Regarding supervisor behavior, a study among psychiatric trainees showed that those with one primary supervisor had higher levels of satisfaction than those with two primary supervisors [24], which may further indicate the importance of consistency in supervision approaches. In another study, positive framed feedback was correlated with greater self-efficacy and better performance [25]. Additionally, the credibility of feedback seems to influence trainees’ satisfaction [26].

The WEMWBS score demonstrated high internal consistency with Cronbach’s alpha, indicating the validity of this tool for evaluating mental well-being among Saudi residents. Similarly, the D-RECT subscales were reliable, with Cronbach’s alpha values and overall good to excellent consistency, allowing factual assessment of clinical supervision across physicians. This finding indicates the relevance of using the WEMWBS and D-RECT in the present study.

Consistent with our results, in a survey of 1278 residents representing 26 specialties, D-RECT was shown to be a valid and reliable instrument for assessing clinical learning quality [4]. The 50-item D-RECT and 11 subscales used in the study included feedback, supervision, patient handover, and professional relations between attendings and others; provided an interactive evaluative approach on what should be maintained; and provided insight into what could be improved in the learning experience based on the trainees’ opinions. In a larger study, a shorter version of the D-RECT was used based on 35 items highlighting nine domains (teamwork, role of specialty tutor, coaching and assessment, formal education, resident peer collaboration, work is adapted to residents’ competence, patient sign-out, educational atmosphere, and accessibility of supervisors) to measure the quality of the learning climate among residents with good reliability outcomes [27]. The other aspect of the study showed that fewer residents were required to complete learning performance evaluations.

The validity of the WEMWBS score for evaluating physicians’ well-being has still not been fully explored. Many barriers to effective clinical supervision include hindering the adequate exchange of feedback between supervisors and supervisees, such as the lack of direct observation of tasks executed by trainees, the desire of tutors to avoid upsetting students with honest criticism when the performance of trainees is below expectations, and the lack of external feedback, which may lead to frequently incorrect self-assessment by learners [23]. In a clinical training environment, unapproachable attendings, clinical work-related time pressure, and discomfort from giving negative feedback affect the feedback process [28]. A survey from the United Kingdom revealed that more than half of the surveyed consultant supervisors never received regular feedback regarding the educational and clinical supervision they gave, and few of them reported receiving such feedback annually [29].

In addition, consultants may face difficulties in providing the optimum supervision quality to their trainees. According to a survey of consultants, their teaching efforts were not financially compensated, and for half of them, teaching medical students was not a part of their job plans. Moreover, the majority of the respondents did not have proper time for teaching [30]. In fact, many supervisors do not have the needed competence and skills to provide effective supervision [31]. Unfamiliarity with professional guidelines, being unaware of the role and responsibilities of a supervisor, being unaware of ethical standards, and inadequate educational preparation were associated with reduced supervision quality [32]. Insufficient supervisor competence may result in an inappropriate education process marked by intolerance, blaming, and inflexibility with supervisees; inability to deal with unmotivated people; inability to manage diversity in trainees’ personalities; inability to share feelings and give adequate feedback; and inability to experience empathy related to personal issues [33].

## 5. Implications and Conclusions

It is important to note that plentiful studies on mental burnout among physicians have been conducted, in contrast to studies on well-being, which are remarkably rare despite the substantial need for effective strategies to potentiate mental well-being by addressing negative factors and prompting positive ones [33]. Despite the valuable insights provided by this study, several limitations must be considered. The relatively small sample size of 64 participants may limit the generalizability of the findings to all otolaryngology residents in Saudi Arabia. Additionally, the use of self-reported questionnaires introduces the potential for response bias, as participants may have provided socially desirable answers rather than fully accurate reflections of their experiences. The four items we used from the D-RECT score can be used in mental health promotion programs to improve physicians’ satisfaction and academic experience. Furthermore, professional coaching programs have some evidence of effectiveness in relieving burnout levels in physicians [20,34].

Clinical supervision and mentorship have a strong impact on physicians’ mental well-being, specifically by promoting self-esteem and a sense of utility at work in the context of medical training. Trained physicians should benefit from frequent feedback sessions where they can freely express themselves and their struggles. Seniors should also be encouraged to reveal what they need in terms of empathetic support and ask for advice on how to cope with stress and emotional exhaustion associated with clinical duties. Based on the study’s findings, residency programs should implement structured supervision models that emphasize regular, constructive feedback and foster positive consultant attitudes to create a supportive learning environment. Incorporating formal mentorship programs and peer support groups can further strengthen the emotional resilience of residents and improve mental well-being. Residency programs should also periodically assess the quality of clinical supervision using validated tools such as the D-RECT to identify areas for improvement and ensure continuous support for trainees. Future research should explore the long-term impact of these interventions and investigate culturally tailored approaches to enhance resident well-being in diverse clinical settings.

## Figures and Tables

**Table 1 healthcare-13-00328-t001:** Patterns of answers to questions about mental well-being.

Item	Frequency, N (%)					Score	
	None of the Time	Rarely	Some of the Time	Often	All of the Time	Mean	SD
I’ve been feeling optimistic about the future	5 (7.8)	10 (15.6)	18 (28.1)	26 (40.6)	5 (7.8)	3.25	1.07
I’ve been feeling useful	5 (7.8)	8 (12.5)	16 (25.0)	18 (28.1)	17 (26.6)	3.53	1.23
I’ve been feeling relaxed	12 (18.8)	12 (18.8)	22 (34.4)	14 (21.9)	4 (6.3)	2.78	1.17
I’ve been feeling interested in other people	7 (10.9)	13 (20.3)	22 (34.4)	14 (21.9)	8 (12.5)	3.05	1.17
I’ve had energy to spare	9 (14.1)	12 (18.8)	18 (28.1)	18 (28.1)	7 (10.9)	3.03	1.22
I’ve been dealing with problems well	3 (4.7)	16 (25.0)	13 (20.3)	25 (39.1)	7 (10.9)	3.27	1.10
I’ve been thinking clearly	4 (6.3)	9 (14.1)	26 (40.6)	18 (28.1)	7 (10.9)	3.23	1.03
I’ve been feeling good about myself	6 (9.4)	14 (21.9)	23 (35.9)	12 (18.8)	9 (14.1)	3.06	1.17
I’ve been feeling close to other people	5 (7.8)	13 (20.3)	22 (34.4)	15 (23.4)	9 (14.1)	3.16	1.14
I’ve been feeling confident	7 (10.9)	12 (18.8)	18 28.1)	18 (21.1)	9 (14.1)	3.16	1.21
I’ve been able to make up my own mind about things	4 (6.3)	9 (14.1)	25 (39.1)	19 (29.7)	7 (10.9)	3.25	1.04
I’ve been feeling loved	9 (14.1)	10 (15.6)	17 (26.6)	19 (29.7)	9 (14.1)	3.14	1.26
I’ve been interested in new things	4 (6.3)	12 (18.8)	20 (31.3)	18 (28.1)	10 (15.6)	3.28	1.13
I’ve been feeling cheerful	4 (6.3)	14 (21.9)	18 (28.1)	21 (32.8)	7 (10.9)	3.20	1.10

**Table 2 healthcare-13-00328-t002:** Internal consistency and descriptive statistics of the different study scales.

Scale	No. Items	Cronbach’s Alpha	Reliability	Mean	SD	Range
WEMWBS	14	0.958	Excellent	44.39	12.95	15–68
Supervision	3	0.740	Good	3.49	0.84	1.33–5.00
Feedback	3	0.829	High	2.95	0.95	1.00–5.00
Coaching assessment	8	0.912	Excellent	3.21	0.86	1.00–4.88
Consultant’s attitude	8	0.928	Excellent	3.57	0.88	1.25–5.00

WEMWBS: Warwick–Edinburgh Mental Well-being Scale.

**Table 3 healthcare-13-00328-t003:** Levels of mental well-being by residency level.

Item	Junior Residents (2nd and 3rd Year)		Senior Residents (4th and 5th Year)		*p*-Value
	Mean	SD	Mean	SD	
I’ve been feeling optimistic about the future	3.19	0.87	3.28	1.16	0.758
I’ve been feeling useful	3.38	1.20	3.60	1.26	0.500
I’ve been feeling relaxed	2.48	1.12	2.93	1.18	0.148
I’ve been feeling interested in other people	2.81	1.17	3.16	1.17	0.262
I’ve had energy to spare	3.10	1.09	3.00	1.29	0.772
I’ve been dealing with problems well	3.14	1.15	3.33	1.08	0.538
I’ve been thinking clearly	3.05	0.80	3.33	1.13	0.317
I’ve been feeling good about myself	2.67	0.73	3.26	1.29	0.057
I’ve been feeling close to other people	2.81	1.12	3.33	1.13	0.090
I’ve been feeling confident	2.90	0.89	3.28	1.33	0.249
I’ve been able to make up my own mind about things	2.95	0.74	3.40	1.14	0.110
I’ve been feeling loved	2.81	1.21	3.30	1.26	0.143
I’ve been interested in new things	3.24	1.14	3.30	1.15	0.833
I’ve been feeling cheerful	2.81	0.98	3.40	1.12	0.045 *
WEMWBS score	41.33	10.43	45.88	13.88	0.189

The values are the means (SDs) of the WEMWBS item scores. WEMWBS: Warwick–Edinburgh Mental Well-being Scale * Statistically significant result (*p* < 0.05).

**Table 4 healthcare-13-00328-t004:** Association of residents’ mental well-being with postgraduate year and perceived clinical supervision.

Factor	Level	WEMWBS Score		*p*-Value
		Mean	SD	
Postgraduate year	2	40.00	8.43	
	3	42.55	12.26	
	4	40.94	12.41	
	5	48.81	14.09	0.127
Supervision	Unsatisfactory (score ≤ 3)	37.58	9.91	
	Satisfactory (score > 3)	48.48	12.94	0.001 *
Feedback	Unsatisfactory (score ≤ 3)	39.42	12.45	
	Satisfactory (score > 3)	51.65	10.04	<0.001 *
Coaching assessment	Unsatisfactory (score ≤ 3)	36.84	12.14	
	Satisfactory (score > 3)	49.23	11.11	<0.001 *
Consultant’s attitude	Unsatisfactory (score ≤ 3)	30.79	11.42	
	Satisfactory (score > 3)	48.20	10.64	<0.001 *

WEMWBS: Warwick–Edinburgh Mental Well-being Scale. * Statistically significant result (*p* < 0.05).

**Table 5 healthcare-13-00328-t005:** Independent factors of mental well-being (multivariate linear regression).

Predictor	B	95%CI		*p*-Value	Model Goodness-of-Fit (Adjusted R^2^)
Enter Method					
(Constant)	28.89	23.23	34.55	<0.001 *	0.387
Supervision	4.49	−1.34	10.32	0.129	
Feedback	6.44	0.49	12.39	0.034 *	
Coaching assessment	2.16	−4.72	9.04	0.532	
Consultant’s attitude	11.21	3.46	18.96	0.005 *	
Stepwise Method					
(Constant)	30.22	24.73	35.70	<0.001 *	0.375
Consultant’s attitude	13.99	7.36	20.63	<0.001 *	
Feedback	7.98	2.40	13.57	0.006 *	

Dependent variable: WEMWBS score. * Statistically significant result (*p* < 0.05).

## Data Availability

Data is contained within the article.

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
