# Peer review of "Exploring the Relationship Between Clinical Supervision and Well-Being in the Otolaryngology Residency Board in Saudi Arabia"

_healthcare, 2025, doi:10.3390/healthcare13030328_

Round 1

Reviewer 1 Report

Comments and Suggestions for Authors

Dear Authors,

please answer my observation point-by-point.

1.     Please list abbreviation  WEMWBS in brackets after first mentioning Warwick-Edinburgh Mental Well-being Scale questionnaire

2.     Was the Dutch Residents Educational Climate Test validated for use in Saudi Arabia?

3.     What about the Warwick-Edinburgh Mental Well-being Scale? Was that validated for the study population?

4.     Translation of the D-RECT was performed how?

5.     What about the translation for the Warwick-Edinburgh Mental Well-being Scale

6.     Line 40: after the word ”performed” there is the number one listed.

7.     Can You add a citation after the statement ” Such work-related factors could affect mental wellbeing and ultimately lead to burnout or mental illness, such as depression or anxiety disorders.”?

8.     Please rephrase the following ”Thus, appropriate supervision in the 56

healthcare setting results from the antecedents and consequences of job well-being.” – the meaning seems lost to me.

9.     What do You mean by ”who were in levels two to five”? – as you later mention 3rd year and 5th year. Also, please explain what KSA stands for.

10.  ”sixty-four postgraduate residents replied to the questionnaire,” – please adapt text, as it seems unfinished

11.  When was the research performed?

12.  Please remote no. 35 from reference list, as it there no reference listed.

Author Response

Please list abbreviation  WEMWBS in brackets after first mentioning Warwick-Edinburgh Mental Well-being Scale questionnaire.

Thank you for the comment the editing done on line 37

Was the Dutch Residents Educational Climate Test validated for use in Saudi Arabia?

Thank you for highlighting this important question. The Dutch Residents Educational Climate Test (D-RECT) has not been formally validated in Saudi Arabia. However, our residency programs are conducted in English and follow international standards, which align closely with the contexts where the D-RECT has been validated, such as in Spain, the Philippines, Denmark, and France. These validations have consistently demonstrated the tool's reliability and adaptability across diverse cultural and linguistic settings.

In our study, we calculated the internal consistency of the D-RECT subscales, which yielded good to excellent Cronbach's alpha values (e.g., Supervision = 0.740, Feedback = 0.829, Coaching Assessment = 0.912, Consultant Attitude = 0.928). This suggests that the tool reliably captures the educational climate in our setting, even without formal local validation.

What about the Warwick-Edinburgh Mental Well-being Scale? Was that validated for the study population?

To our knowledge, the WEMWBS has not been formally validated specifically for the Saudi population. However, it has been extensively used and validated in diverse international contexts, including healthcare settings, with robust psychometric properties.

In our study, we calculated the internal consistency of the WEMWBS among our participants, which yielded a Cronbach’s alpha of 0.958. This demonstrates excellent reliability for the scale within our study population. Additionally, the scale's broad focus on positively worded statements and psychological functioning makes it versatile and widely applicable across different cultural settings.

Given the English-language foundation of our residency program and the international nature of the WEMWBS, we deemed it appropriate for assessing mental well-being in our participants.

Translation of the D-RECT was performed how?

What about the translation for the Warwick-Edinburgh Mental Well-being Scale

Thank you for your). In our study, we utilized its original English form without translation. This approach was justified because the residency training programs in Saudi Arabia are conducted in English, and all participants were fluent in the language. Therefore, no linguistic

Line 40: after the word ”performed” there is the number one listed.

Thank you for pointing this out. Upon reviewing the manuscript, we could not locate the number "1" after the word "performed" on line 40 as mentioned in the comment. It is possible that this was a formatting artifact or a typographical error during the manuscript's upload or conversion process

Can You add a citation after the statement ” Such work-related factors could affect mental wellbeing and ultimately lead to burnout or mental illness, such as depression or anxiety disorders.”?

Thank you for the comment the editing done.

Please rephrase the following ”Thus, appropriate supervision in the 56

healthcare setting results from the antecedents and consequences of job well-being.” – the meaning seems lost to me

Thank you for the comment the editing done.

What do You mean by ”who were in levels two to five”? – as you later mention 3rd year and 5th year. Also, please explain what KSA stands for

Thank you for the comment the editing done.

sixty-four postgraduate residents replied to the questionnaire,” – please adapt text, as it seems unfinished

Thank you for the comment the editing done.

When was the research performed?

Thank you for the comment the editing done. (data were collected over an 8-week period from June to August 2022.)

Please remote no. 35 from reference list, as it there no reference listed

Thank you for the comment the editing done.

Reviewer 2 Report

Comments and Suggestions for Authors

Evaluation of the manuscript “Exploring the relationship between Clinical Supervision and Well-being in Otolaryngology Residency Board in Saudi Arabia” submitted to the journal Healthcare

The study investigates the relationship between clinical supervision and mental well-being among otolaryngology residents in Saudi Arabia, aiming to understand how different levels of supervision impact well-being during training. An analytical cross-sectional study was conducted with 64 residents from the second to the fifth year, using the Dutch Residents Educational Climate Test (D-RECT) to assess clinical supervision and the Warwick-Edinburgh Mental Well-being Scale (WEMWBS) to measure mental well-being. The results showed that higher satisfaction levels with supervision, feedback, assessment, and consultant’s attitude were significantly associated with better resident well-being scores. However, no statistically significant differences were observed between the well-being of junior and senior residents.

The research highlights the importance of clinical supervision as a key resource for promoting residents' well-being, particularly through positive feedback and supportive supervisor attitudes. The findings suggest that investing in high-quality supervision can enhance both residents' performance and the learning environment.

The study addresses a relevant topic in medical education, with the potential to impact the training of healthcare professionals. The use of validated instruments to assess clinical supervision and mental well-being ensures data reliability. However, the cross-sectional design limits causal inferences. Additionally, the lack of detailed information on variations between training centers, combined with restricted reflection on methodological limitations and confounding variables such as workload and organizational structure, represents a limitation. Despite these issues, the study provides valuable evidence on the relationship between clinical supervision and residents' well-being, which can inform future research and improvements in medical residency programs.

In this context, several points could be improved to strengthen the manuscript:

ABSTRACT: The authors focused on contextualizing the problem but did not provide enough information about the study's results and conclusions. A comprehensive revision of the abstract is recommended, with greater emphasis on the core aspects of the research (objective, method, results, and conclusions/contributions).

INTRODUCTION: While the introduction effectively contextualizes the problem and discusses the importance of mental well-being and clinical supervision in medical residency, the final paragraph requires revision. The paragraph appears to mix the study's objectives, contributions, and implications, which can obscure the article's purpose. It is recommended to clearly and directly state the study's main objective at the beginning of the paragraph, while addressing contributions, implications, or conclusions in the discussion or conclusion sections. Furthermore the introduction could be more concise in some sections, maintaining a clear focus on the study's justification and the gaps it aims to address.

METHODS: Although the methods are described in general terms, they lack sufficient detail to ensure the study's reproducibility. It is recommended to clarify: the total population; the number of participating centers and whether the data collection process was consistent across them; how many first-year residents were excluded; the response rate and how non-responses were handled. Additionally, were the instruments used validated for the cultural context of Saudi Arabia? Finally, potential biases in data collection should be addressed.

DISCUSSION: The discussion effectively relates the findings to the existing literature but could benefit from a more in-depth examination of the study's limitations, such as sample size, self-report bias, and confounding variables (if addressed, these should be specified in the methods). The discussion could also elaborate on the practical implications of the findings and provide clearer directions for future research.

CONCLUSION: The conclusion should include specific recommendations for improving medical residency programs and implementing interventions based on the study's findings.

Author Response

Reviwer 2:

The authors focused on contextualizing the problem but did not provide enough information about the study's results and conclusions. A comprehensive revision of the abstract is recommended, with greater emphasis on the core aspects of the research (objective, method, results, and conclusions/contributions)

Thank you for the comment the editing done and abstract reformulated.

While the introduction effectively contextualizes the problem and discusses the importance of mental well-being and clinical supervision in medical residency, the final paragraph requires revision. The paragraph appears to mix the study's objectives, contributions, and implications, which can obscure the article's purpose. It is recommended to clearly and directly state the study's main objective at the beginning of the paragraph, while addressing contributions, implications, or conclusions in the discussion or conclusion sections. Furthermore the introduction could be more concise in some sections, maintaining a clear focus on the study's justification and the gaps it aims to address

Thank you for the comment the editing done and abstract reformulated.

Although the methods are described in general terms, they lack sufficient detail to ensure the study's reproducibility. It is recommended to clarify: the total population; the number of participating centers and whether the data collection process was consistent across them; how many first-year residents were excluded; the response rate and how non-responses were handled. Additionally, were the instruments used validated for the cultural context of Saudi Arabia? Finally, potential biases in data collection should be addressed

Thank you for the comment , The study targeted all otolaryngology residents in Saudi Arabia’s national residency program, specifically those in their 2nd to 5th years, to ensure adequate exposure to clinical supervision across regional centers. First-year residents were excluded due to their limited rotation duration with most of rotation not at otolaryngology department, and data were collected through a standardized online survey distributed by chief residents and program directors across all centers. A total of 64 residents participated, with non-responses excluded from the analysis. Although the Dutch Residents Educational Climate Test (D-RECT) and Warwick-Edinburgh Mental Well-being Scale (WEMWBS) were not specifically validated for the Saudi context, their internal consistency was confirmed, with Cronbach’s alpha values exceeding 0.7, indicating good reliability. Potential biases, including self-selection and reliance on self-reported data, were acknowledged, and the limitations of non-response bias were addressed in the discussion.

The discussion effectively relates the findings to the existing literature but could benefit from a more in-depth examination of the study's limitations, such as sample size, self-report bias, and confounding variables (if addressed, these should be specified in the methods). The discussion could also elaborate on the practical implications of the findings and provide clearer directions for future research.

Thank you for the comment the editing done and imported at Implications section.

The conclusion should include specific recommendations for improving medical residency programs and implementing interventions based on the study's findings

Thank you for the comment the editing done and conclusion reformulated.

Reviewer 3 Report

Comments and Suggestions for Authors

Thank you having me as a reviewer - here are the areas for improvement in the manuscript:

  1. Introduction

    • The introduction could provide more recent references and a broader contextual background to enhance the novelty and relevance of the study.
    • Include recent studies on resident well-being and clinical supervision from diverse geographic contexts to strengthen the introduction - 
    • DOI: 10.1186/s12909-023-04867-0
  2. Methods Section

    • The description of the research design and sampling technique needs more clarity, particularly regarding the rationale for selecting specific residency levels.
    • Provide a detailed explanation of the sampling strategy and justify the exclusion criteria to avoid potential bias.
  3. Results Presentation

    • The results are presented clearly, but the manuscript would benefit from more visual aids, such as graphs or charts, to make data interpretation easier.
    • Include more visual elements to highlight key findings, making the results more accessible to readers.
  4. Discussion Section

    • The discussion section could benefit from a more critical analysis of the findings in relation to existing literature.
    • Strengthen the discussion by critically comparing the study’s findings with previous studies and elaborating on potential implications for practice and policy.
  5. Conclusion

    • The conclusion is aligned with the results but could provide more actionable recommendations for future research and practice.
    • Include specific recommendations for healthcare institutions and residency programs on how to enhance clinical supervision to improve resident well-being.
  6. Language and Grammar

    • There are minor language and grammar issues throughout the manuscript that could be polished to improve readability.
    • Conduct a thorough proofreading or consider professional editing to ensure linguistic accuracy and clarity.

Author Response

  • The introduction could provide more recent references and a broader contextual background to enhance the novelty and relevance of the study.
  • Include recent studies on resident well-being and clinical supervision from diverse geographic contexts to strengthen the introduction - DOI: 10.1186/s12909-023-04867-0
  • Thank you for your valuable feedback and for suggesting ways to strengthen the introduction. In response, we have revised the introduction to provide a broader contextual background and included more recent and relevant references to enhance the study's novelty and relevance.
  • Regarding the suggested study (DOI: 10.1186/s12909-023-04867-0), we appreciate your recommendation. However, upon review, we found that this study primarily focuses on burnout, which, although related, differs from our study’s main focus on mental well-being. Our research emphasizes the positive dimensions of mental health and overall well-being rather than the presence or impact of burnout.
  • We are grateful for your thoughtful suggestions and believe these revisions have significantly strengthened the introduction

  • The description of the research design and sampling technique needs more clarity, particularly regarding the rationale for selecting specific residency levels. Provide a detailed explanation of the sampling strategy and justify the exclusion criteria to avoid potential bias

Thank you for the comment the editing done and methodology explained and  reformulated. The study targeted all otolaryngology residents in Saudi Arabia’s national residency program, specifically those in their 2nd to 5th years, to ensure adequate exposure to clinical supervision across regional centers. First-year residents were excluded due to their limited rotation duration with most of rotation not at otolaryngology department.

Results and Discussion :

Thank you for your constructive feedback on the presentation of the results and the discussion section. We appreciate your valuable suggestions and have made revisions to the Implications section to enhance the accessibility of the results while maintaining detailed statistical data for readers interested in in-depth analysis. This approach balances clarity and comprehensiveness, aligning with the study’s structure as part of a master’s thesis.

Additionally, the Discussion section has been strengthened to emphasize the importance of using validated tools like the D-RECT and WEMWBS in assessing clinical supervision and mental well-being. We highlighted how the depth and reliability of these tools provide actionable insights that can be effectively applied within the Saudi Arabian medical education context

We believe these improvements have enhanced the clarity and impact of the manuscript. Thank you again for your insightful feedback, which has contributed significantly to refining our work

Conclusion

  • The conclusion is aligned with the results but could provide more actionable recommendations for future research and practice. Include specific recommendations for healthcare institutions and residency programs on how to enhance clinical supervision to improve resident well-being
  • Thank you for the comment the editing done and conclusion reformulated.

Language and Grammar:

Thank you for the comment and professional editing has been conducted to improve the clarity and quality of the manuscript.

Round 2

Reviewer 2 Report

Comments and Suggestions for Authors

Second evaluation of the manuscript “Exploring the relationship between Clinical Supervision and Well-being in Otolaryngology Residency Board in Saudi Arabia”, submitted to the journal Healthcare.

The authors present a revised version of the manuscript, incorporating adjustments based on the recommendations provided in the first review.

The abstract has been reformulated to include the central aspects of the research, such as the objective, method, results, and conclusions. Additionally, it now presents the main findings (e.g., a positive association between clinical supervision and well-being) and highlights the practical implications. This reformulation satisfactorily addresses the recommendations.

The introduction contextualizes the problem and emphasizes the importance of mental well-being and clinical supervision. Although the final paragraph still combines objectives, contributions, and implications, the objective is explicitly stated in the first sentence. Therefore, the rewording meets the recommendations.

The methods section has been expanded to address the reviewers’ concerns. The target population has been clearly defined, and the data collection process, including exclusions and response rates, is described in detail. The reliability of the instruments (D-RECT and WEMWBS) is reported, although these tools lack specific validation for the Saudi context. The authors acknowledged the potential for bias and included limitations such as non-response bias. This section has been significantly improved, satisfactorily addressing the recommendations.

The discussion connects the findings to the existing literature, and the practical implications have been elaborated upon, including suggestions for future interventions.

The conclusion has been reformulated and now includes specific recommendations for improving medical residency programs, such as structured feedback and positive attitudes from consultants, fully addressing the reviewers’ requests.

Considering the substantial improvements made to the manuscript following the authors' revisions, I recommend it for publication.

Author Response

Dear Reviewer, 
We sincerely thank the reviewer for the thorough evaluation and constructive feedback on our manuscript titled “Exploring the relationship between Clinical Supervision and Well-being in Otolaryngology Residency Board in Saudi Arabia.” We appreciate the acknowledgment of the substantial improvements made to the manuscript and provide the following detailed point-by-point responses to address the comments:

  • Reviewer’s Comment 1: The authors present a revised version of the manuscript, incorporating adjustments based on the recommendations provided in the first review.
  • Response: Thank you for recognizing the efforts made in revising the manuscript. We have carefully addressed all recommendations provided in the first review to enhance the clarity and quality of the manuscript.
  • Reviewer’s Comment 2: The abstract has been reformulated to include the objective, method, results, and conclusions, highlighting the main findings and practical implications. The reformulation satisfactorily addresses the recommendations.
  • Response: We appreciate the positive feedback on the revised abstract. The abstract now succinctly outlines the study’s core elements and emphasizes its significance, as requested.
  • Reviewer’s Comment 3: The introduction contextualizes the problem and emphasizes the importance of mental well-being and clinical supervision. The final paragraph combines objectives, contributions, and implications, but the objective is explicitly stated. The rewording meets the recommendations.
  • Response: Thank you for acknowledging the improvements in the introduction. While the final paragraph combines multiple elements, we ensured the primary objective is clearly articulated in the first sentence, aligning with the reviewers’ feedback.
  • Reviewer’s Comment 4: The methods section has been expanded, addressing concerns about the target population, data collection process, and reliability of instruments. The authors acknowledge potential biases and limitations. This section has been significantly improved.
  • Response: We are pleased that the expanded methods section meets your expectations. The revisions detail the target population, exclusions, response rates, and limitations, including potential biases. Additionally, while we note that the instruments (D-RECT and WEMWBS) lack specific validation in the Saudi context, we have transparently acknowledged this limitation in the manuscript.
  • Reviewer’s Comment 5: The discussion connects findings to the literature, and practical implications have been elaborated upon, including suggestions for future interventions.
  • Response: Thank you for acknowledging the strengthened discussion section. We ensured the findings were contextualized within existing literature, and practical implications were clearly articulated. Suggestions for future interventions, including enhancing clinical supervision and well-being in residency programs, were highlighted.
  • Reviewer’s Comment 6: The conclusion includes specific recommendations for improving medical residency programs, fully addressing the reviewers’ requests.
  • Response: We are glad the reformulated conclusion meets your expectations. Specific recommendations, such as structured feedback and fostering positive attitudes among consultants, were integrated to provide actionable insights for residency program improvements.
  • Reviewer’s Comment 7: Considering the substantial improvements, the manuscript is recommended for publication.
  • Response: We deeply appreciate the acknowledgment of our efforts and the recommendation for publication. Your constructive feedback has been invaluable in enhancing the manuscript.